# Thin Delivery Stents Can Obviate the Need for Additional Fistula Dilatation of Large Diameter in Endoscopic Ultrasound-Guided Hepaticogastrostomy

**DOI:** 10.3390/jcm13216328

**Published:** 2024-10-23

**Authors:** Tomoki Ogata, Yusuke Kurita, Takamitsu Sato, Shin Yagi, Sho Hasegawa, Kunihiro Hosono, Noritoshi Kobayashi, Itaru Endo, Kensuke Kubota, Atsushi Nakajima

**Affiliations:** 1Department of Gastroenterology and Hepatology, Yokohama City University, Yokohama 236-0004, Japan; tomo1993baske@gmail.com (T.O.); tkmtsato@yokohama-cu.ac.jp (T.S.); e103083b@gmail.com (S.Y.); t166064d@yokohama-cu.ac.jp (S.H.); hiro1017@yokohama-cu.ac.jp (K.H.); kubotak@yokohama-cu.ac.jp (K.K.); nakajima-tky@umin.ac.jp (A.N.); 2Department of Oncology, Yokohama City University, Yokohama 236-0004, Japan; norikoba@yokohama-cu.ac.jp; 3Department of Gastroenterological Surgery, Yokohama City University Hospital, Yokohama 236-0004, Japan; endoit@yokohama-cu.ac.jp

**Keywords:** biliary obstruction, endoscopic ultrasound hepaticogastrostomy, thin delivery stents, fistula dilation

## Abstract

**Background/Objectives:** Endoscopic ultrasound-guided hepaticogastrostomy (EUS-HGS) often requires fistula dilation owing to the placement of a large diameter of the delivery stent. The recently developed delivery devices, which are as thin as 5.9/6.0 Fr, may save the need for fistula dilation. Therefore, we investigated whether large fistula dilation would be required or not in the case of this newly developed thin-diameter delivery stents. **Methods:** We conducted a retrospective study involving 33 patients implemented with a self-expandable metal stent (SEMS) during EUS-HGS. The patients were categorized based on the delivery device diameter into thin (n = 13; delivery device diameter: 5.9/6.0 Fr) and thick (n = 20; delivery device diameter: 8.5 Fr) groups. We compared the initial rate of success, technical success, and clinical success between the thin and thick groups. The initial rate of success was defined as successful stent placement without a balloon or large diameter mechanical dilation. **Results:** The rate of the initial stenting success was significantly higher in the thin group (100% [13/13]) compared with that in the thick group (65.0% [13/20]) (*p* = 0.027). In the thick group, seven cases with technical difficulty in terms of stent placement could be successfully completed with additional fistula dilation with a 9 Fr bougie dilator or 4 mm balloon dilator; this resulted in a technical success of 100% in both groups ultimately. The rate of clinical success was 100% and 95.0% in the thin and thick groups, respectively (*p* = 1.00). **Conclusions:** Thin delivery stents may facilitate stent placement without the need for a balloon fistula or large-diameter mechanical dilation.

## 1. Introduction

Recently, endoscopic ultrasound-guided hepaticogastrostomy (EUS-HGS) has been performed for biliary obstruction in cases of failed transpapillary drainage in recent years [1,2,3,4,5]. The safety of EUS-HGS has been previously reported. However, some adverse events can still take place [6,7,8,9,10,11,12]. EUS-HGS is a highly technical procedure and should only be performed by highly skilled endoscopists at high-volume centers. EUS-HGS is composed of various steps to ensure its success; fistula dilation is one important technique, as some patients have difficulty with dilating the stomach wall or intrahepatic bile duct [6,9,13,14].

Notably, conventional metal stents may require balloon fistula dilation or mechanical dilation of a large diameter in addition to mechanical dilation to implant an 8.5 Fr self-expandable metal stent (SEMS) with such a thick delivery [15]. On the other hand, the use of thin delivery stents (5.9 Fr or 6 Fr) is expected to make it easier for the endoscopist to insert the stent [16,17]. Although there have been cases where the thick delivery stent required balloon fistula dilation or 9 Fr mechanical dilation, thin delivery stents may be inserted without the need for additional dilation. However, a comparison of balloon fistula or mechanical dilation of a large diameter before insertion in thin- and thick-diameter delivery stents has not been fully carried out. Therefore, we compared the need for large fistula dilation using thin and thick delivery stents.

## 2. Materials and Methods

### 2.1. Patients

This is a retrospective cohort case study. The patients who underwent EUS-HGS were within the Yokohama City University Hospital between April 2015 and January 2023. The main indications for EUS-HGS at our hospital are that it is difficult to reach the duodenal papilla due to reconstructive bowel or gastrointestinal stenosis, bile duct cannulation is difficult, and cholangitis is poorly controlled with transpapillary drainage with ERCP alone. The inclusion criteria of the present study were 1) patients who had a metal stent implanted at the time of EUS-HGS, and the exclusion criteria of the present study were 1) patients who had a plastic stent implanted and 2) patients who had balloon fistula dilatation, or a larger-diameter bougie catheter (≥9 Fr) before initial metal stenting. For comparison, patients were divided into thin delivery stent (thin group) and thick delivery stent (thick group) groups based on the stent delivery system at our institution (Figure 1). EUS-HGS is indicated when bile duct intubation is impossible or when reaching the duodenal papilla is not possible due to the obstruction of the gastric or surgical anatomical changes outlet.

Ethics approval for this study and the waiver of the need for consent to participate were approved by the Institutional Review Board of Yokohama City University Hospital (B2006003, approved 1 May 2020). In this retrospective observational study, only medical information without invasion to participants is used.

### 2.2. Procedures of EUS-HGS

EUS-HGS was performed in all cases by experienced endoscopists who specialized in EUS-guided procedures. The standard procedure of EUS-HGS used at our hospital is as follows: The endoscopic ultrasound (EU-ME2 or EU-ME1; Olympus, Tokyo, Japan, and GF-UCT260) was performed from the stomach to show the left intrahepatic bile duct (segment 2: B2 or segment 3: B3). A 19 G needle or a 22 G needle (Sono Tip Pro Control; Medi-Globe GmbH, Rosenheim, Germany; Expect; Boston Scientific, Boston, MA, USA; or EZ shot 3 plus; Olympus, Tokyo, Japan) was used to puncture the left intrahepatic bile duct. After puncturing the intrahepatic bile duct, and aspirating bile juice, the bile duct was confirmed to be a bile duct by contrast. A 0.025-inch guidewire (VisiGlide2; Olympus, Tokyo, Japan), or a 0.018-inch guidewire (Fielder 18; Asahi Intec, Aichi, Japan), was inserted into the intrahepatic bile duct. Tract dilation from the stomach to bile duct was conducted using a 7 Fr bougie dilation catheter (ES dilator; Zeon Medical Co., Ltd., Tokyo, Japan, or Soehendra Biliary Dilation Catheter; Cook Japan, Tokyo, Japan).

After the tract bougie dilation, SEMS was inserted, with a stent diameter of either 6 mm or 8 mm and a length of 10 cm or 12 cm. The thin delivery stents (thin group) were placed using a delivery device with a diameter of 5.9 Fr (HANAROSTENT Biliary Full Cover Benefit; Boston Scientific, Boston, MA, USA) or 6 Fr (EGIS Biliary Full Cover Stent; SB-KAWASUMI, Kanagawa, Japan) (Figure 2). Thick delivery stents (thick group) were placed using 8.5 Fr delivery stents (Niti-S S-type biliary stents; Taewoong Medical, Seoul, Korea, and HANAROSTENT Biliary Full Cover; Boston Scientific, Boston, MA, USA) (Figure 3).

In cases where SEMS insertion was unsuccessful after initial dilation using a 7 Fr bougie dilation catheter, additional dilatation was performed using a balloon catheter (REN biliary dilation catheter; KANEKA, Osaka, Japan, 4 mm) or a large diameter bougie catheter (Soehendra Biliary Dilation Catheter; 9 Fr in diameter).

### 2.3. Endpoints

The primary endpoint was to compare the rate of initial stenting success between the thin and thick stent groups. Initial stenting success was defined as cases in which stenting was possible with 7 Fr mechanical fistula dilation alone, without balloon dilation of the fistula or a large diameter bougie catheter (<9 Fr). In cases where initial stenting was unsuccessful, additional fistula dilation was conducted using a balloon catheter or a large mechanical catheter (9 Fr) depending on the situation. The secondary endpoints were the rate of technical success, the rate of clinical success, time to recurrent biliary obstruction (TRBO), and adverse events between the thin and thick groups. Technical success was defined as the feasibility of stent placement in the intended bile duct. Clinical success was defined as a successful procedure in which the level of total bilirubin was decreased to normal or ≥50% within 30 days. Recurrent biliary obstruction (RBO) was defined as stent occlusion and migration. TRBO was defined as the period from stent placement to the occurrence of RBO [18]. Adverse events associated with the EUS-HGS were described in accordance with the American Society for Gastrointestinal Endoscopy dictionary within 1 month [19].

### 2.4. Statistical Analysis

The proportions of categorical variables were compared using Fisher’s exact test. The distributions of continuous variables pertaining to the baseline characteristics of the two treatment groups in the cohorts were compared using the Mann–Whitney U test. TRBO is evaluated using the Kaplan–Meier method and the log-rank test. A *p*-value < 0.05 was considered statistically significant. Statistical analyses were performed using JMP 17.0 (SAS Institute Inc., Cary, NC, USA).

## 3. Results

SEMS were placed through EUS-HGS in 84 patients; of these, 33 had SEMS placed without balloon fistula dilatation or a larger-diameter bougie catheter before the initial metal stent implantation in this study. Of the 33 patients, 13 and 20 were in the thin and thick groups, respectively (Figure 1). The characteristics of the patients are presented in Table 1. The age, sex, etiology of biliary obstruction, or location of biliary strictures did not significantly differ between thin and thick groups.

### Outcomes of EUS-HGS

Table 2 shows the outcomes of EUS-HGS. There were significant differences in the initial stenting success rates for SEMS placement between the two groups; the rates were 100% (13/13) and 65.0% (13/20) in the thin and thick groups, respectively (*p* = 0.027). Six of the seven cases who failed initial stent placement had attempted stent placement in B3. The seven cases in the thick group, where stent insertion was a failure, were all successfully stented with 9 Fr mechanical or balloon fistula dilation in the end. The rate of technical success achieved increased to 100% in both groups. The rate of clinical success was 100% in the thin group and 95.0% in the thick group (*p* = 1.00) (Figure 4).

TRBO is shown in Figure 2. The median TRBO for the thin delivery stent group was 86 days (range, 2–199), and for the thick delivery stent group this was 81 days (range, 9–241), meaning they were not significantly different (hazard ratio, 1.52; 95% confidence interval, 0.55–4.23; *p* = 0.42).

Biliary peritonitis was observed as an adverse event after EUS-HGS in 7.7% of patients in the thin group and 5.0% in the thick group; however, these values were not significantly different (*p* = 1.00). No case of other serious complications was observed.

## 4. Discussion

EUS-HGS is one of the biliary drainage techniques. There are limitations to these techniques, such as ensuring a standard technique and the use of safe devices, which could affect its appeal not only for patients but also for endoscopists. Advances in devices have lured endoscopists to the use of devices with thinner outer diameters. However, metal stents require adequate fistula dilation owing to the large external diameter of conventional thick stent delivery devices depending on the case [15]. None of the thin delivery stents used in the present study required large fistula dilation with bougie or balloon dilation. However, thick delivery stents require balloon fistula dilation or 9 Fr mechanical dilation. In this study, we compared the thin and thick delivery stents used for EUS-HGS to elucidate their efficacy for biliary drainage without the additional need for balloon fistula or mechanical dilation. Thin delivery stents may be easier to use and may reduce the need for unnecessary fistula dilation. As a result, this stent would be promising for patients undergoing EUS-HGS.

Most of the cases of the failed initial stenting of thick delivery stents were in the B3 bile duct. This may be because the B3 bile duct has a large scope and stent insertion angle, making stent insertion more complicated. Previous studies have also reported that tight puncture angles require fistula dilation [20]. Balloon fistula dilation or large bougie fistula may be necessary in B3 if a thick delivery stent is to be implanted.

Although TRBO was not significantly different in this study, the thin delivery stent had a better patency duration than the thick delivery stent. Studies on TRBO of thin delivery stents have been reported only in a small number of cases, but previous reports have compared TRBO by delivery diameter and found no significant differences [16,21]. It is possible that even a narrow delivery stent can provide a patency period that is comparable to that of a conventional thick delivery stent. However, the number of cases is small, and the further accumulation of cases is desirable.

The adverse events associated with the use of thin and thick stents implanted using EUS-HGS were not significantly different. EUS-HGS was also safely performed using thin delivery stents in this study. In EUS-HGS procedures, biliary leakage reportedly occurs because various devices must be replaced before stenting [22]. The dilatation step carries with it the risk of bile leaking into the abdominal cavity after dilatation [9,23]. In another study, peritonitis was observed in 3.1% of cases of endoscopic ultrasound-guided biliary drainage with a thick delivery stent, but not with a thin delivery stent [21]. The use of thin delivery stents may be advantageous in reducing unnecessary dilation procedures and avoiding peritonitis.

A lumen-apposing metal stent (LAMS) was used for bile duct drainage recently, which is convenient because it combines a stent and an electrocautery dilator in one unit [24]. Previous reports have shown that LAMS can be used safely during bile duct drainage [25]. However, bleeding has been reported as an adverse event when using an electrocautery dilator [14,26]. In this study, fistula dilation was not required for the placement of a thin delivery stent. Thin delivery stents may be useful because they can be safely performed without the need for energization and balloon fistula dilation.

Our study suffers from the limitation that it was a single-facility retrospective study of a small group of patients. Therefore, prospective randomized controlled trials with many cases are required to validate our findings. We are planning to examine prospective randomized controlled trials with a large number of cases in the future. Also, in this study, all patients in this study underwent fistula dilation with a 7 Fr bougie dilation catheter before the initial stenting. Based on the present study, there is a possibility that thin delivery stents can be used for stenting without fistula dilation, but a further evaluation is desirable.

## 5. Conclusions

Thin delivery stents may facilitate stent placement without the need for balloon fistula dilation or large-diameter mechanical dilation. Thin delivery stents may be a less invasive and safer approach to stent placement.

## Figures and Tables

**Figure 1 jcm-13-06328-f001:**
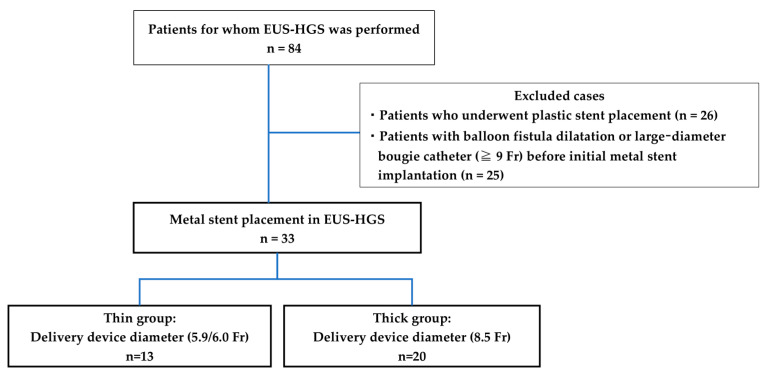
Flowchart of the study selection process.

**Figure 2 jcm-13-06328-f002:**
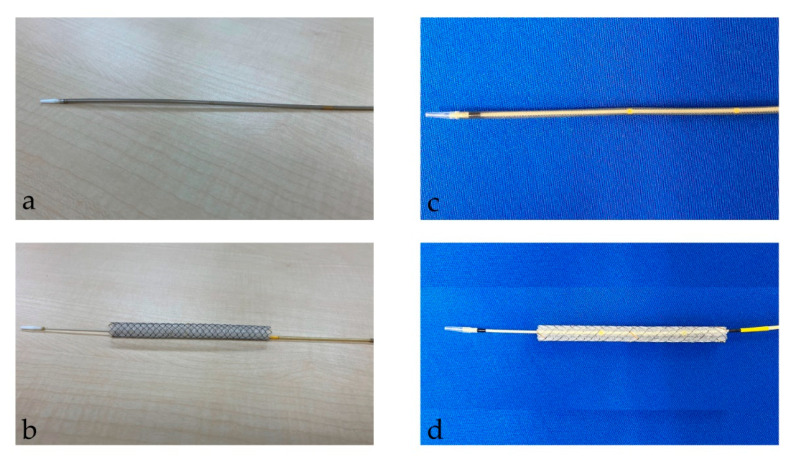
This figure shows a thin delivery stent. (**a**,**b**) shows a stent delivery device with a diameter of 5.9 Fr (HANAROSTENT Biliary Full Cover Benefit; Boston Scientific, Boston, MA, USA). (**a**) shows the stent before deployment; (**b**) shows it after deployment. “© 2024 Boston Scientific Corporation. All rights reserved”. (**c**,**d**) show the stent delivery device with a diameter of 6 Fr (EGIS Biliary Full Cover Stent; SB-KAWASUMI, Kanagawa, Japan). (**c**) shows the stent before deployment; (**d**) shows it after deployment.

**Figure 3 jcm-13-06328-f003:**
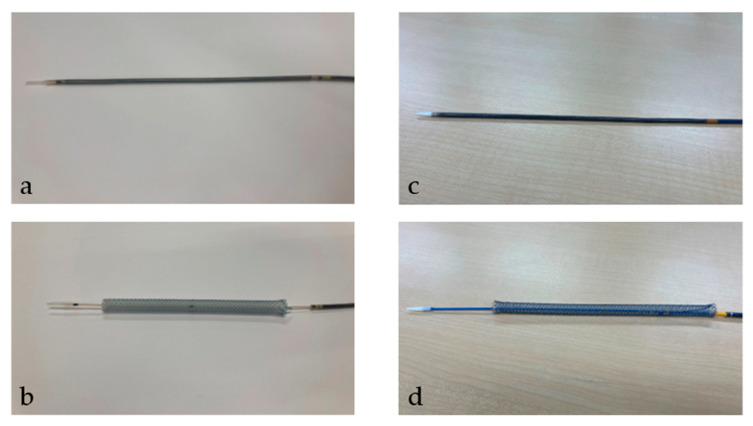
This figure shows a thick delivery stent. Thick delivery stent device with a diameter of 8.5 Fr. (**a**,**b**) show Niti-S S-type biliary stents (Taewoong Medical, Seoul, Korea). (**a**) shows the stent before deployment; (**b**) shows it after deployment. (**c**,**d**) show HANAROSTENT Biliary Full Cover (Boston Scientific, Boston, MA, USA). (**c**) shows the stent before deployment; (**d**) shows it after deployment. “© 2024 Boston Scientific Corporation. All rights reserved”.

**Figure 4 jcm-13-06328-f004:**
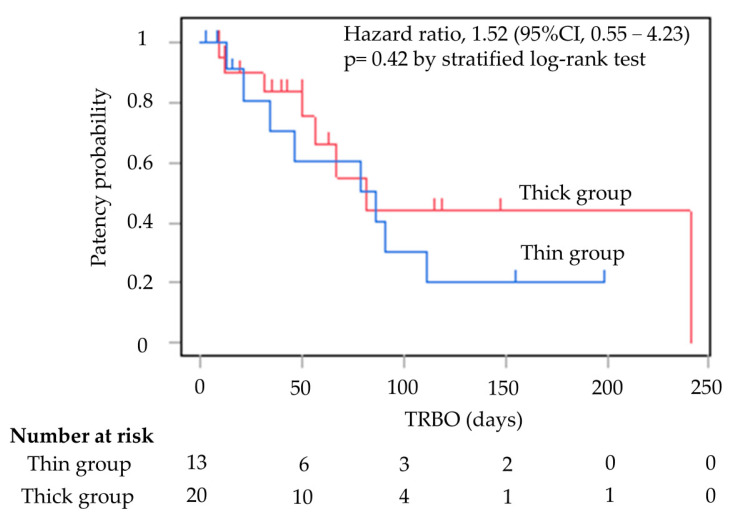
Kaplan–Meier curves of TRBO thin group and thick group. The median TRBO for the thin group was 86 days (range, 2–199). The median TRBO for the thick group was 81 days (range, 9–241). TRBO was not significantly different (hazard ratio, 1.52; 95% confidence interval, 0.55–4.23; *p* = 0.42). TRBO; time to recurrent biliary obstruction.

**Table 1 jcm-13-06328-t001:** The characteristics of patients.

	Thin Group n = 13	Thick Group n = 20	*p*-Value
Age in years, median (range)	77 (45–91)	71.5 (36–83)	0.51
Sex, Male (%)	4 (30.8)	11 (55.0)	0.28
Primary disease (%)			0.74
Pancreatic cancer	6 (46.2)	7 (35.0)	
Cholangiocarcinoma	4 (30.8)	9 (45.0)	
Other	3 (23.1)	4 (20.0)	
Indications for EUS-HGS (%)			0.50
Duodenal stenosis	5 (38.5)	5 (25.0)	
Insufficient papillary drainage	2 (15.4)	6 (30.0)	
Failed papillary cannulation	4 (30.8)	3 (15.0)	
Altered anatomy	1 (7.7)	1 (5.0)	
Other	1 (7.7)	5 (25.0)	
Puncture site (%)			1.00
B2	4 (30.8)	5 (25.0)	
B3	9 (69.2)	15 (75.0)	
Bile duct diameter (mm), median (range)	5.0 (3–9)	5.2 (4–11)	0.54
Fistula dilation device before initial stenting (%)			1.00
Bougie dilator (7 Fr)	13 (100.0)	20 (100.0)	
Stents (%)			N/A
EGIS 6 mm/10 cm (6 Fr)	8 (61.6)	-	
HANAROSTENT Benefit 6 mm/12 cm (5.9 Fr)	5 (38.5)	-	
Niti-s S-type 6 mm/12 cm (8.5 Fr)	-	17 (85.0)	
Niti-s S-type 8 mm/12 cm (8.5 Fr)	-	2 (10.0)	
HANAROSTENT 6 mm/12 cm (8.5 Fr)	-	1 (5.0)	

EUS-HGS; endoscopic ultrasound-guided hepaticogastrostomy.

**Table 2 jcm-13-06328-t002:** Outcomes of EUS-HGS.

	Thin Group n = 13	Thick Group n = 20	*p*-Value
Initial stenting success	100% (13/13)	65.0% (13/20)	0.027
Additional fistula dilation (%)	0 (0.0)	7 (35.0)	
Bougie dilator (9 Fr)	0 (0.0)	3 (15.0)	0.26
Balloon dilator (4 mm)	0 (0.0)	4 (20.0)	0.14
Successful stent placement after additional fistula dilation	-	100% (7/7)	N/A
Treatment time (min), median (range)	30.0 (16–45)	30.0 (17–89)	0.45
Technical success	100% (13/13)	100% (20/20)	1.00
Clinical success	100% (13/13)	95.0% (19/20)	1.00
Adverse events			
Biliary peritonitis	7.7% (1/13)	5.0% (1/20)	1.00
Bleeding	0% (0/13)	0% (0/20)	1.00
Cholangitis	0% (0/13)	0% (0/20)	1.00

EUS-HGS; endoscopic ultrasound-guided hepaticogastrostomy.

## Data Availability

The datasets used and/or analyzed during the current study are available from the corresponding author on reasonable request.

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
