# Peer review of "Thin Delivery Stents Can Obviate the Need for Additional Fistula Dilatation of Large Diameter in Endoscopic Ultrasound-Guided Hepaticogastrostomy"

_jcm, 2024, doi:10.3390/jcm13216328_

Round 1
Reviewer 1 Report
Comments and Suggestions for Authors
This is indeed an interesting finding. The thin stent group shows potential clinical advantages in reducing the opportunity for single instrument exchange, which is very important for improving operational efficiency and reducing patient discomfort. However, as you mentioned, the small sample size may affect the reliability and generalizability of the results.
Have the author considered to provide further image for the compare between thin stent and thick stent with photo in your figure (before deployment and after deployment)
Comments on the Quality of English LanguageMinor revision needed only
Author Response
Comments 1: This is indeed an interesting finding. The thin stent group shows potential clinical advantages in reducing the opportunity for single instrument exchange, which is very important for improving operational efficiency and reducing patient discomfort. However, as you mentioned, the small sample size may affect the reliability and generalizability of the results.
Response 1: Thank you for important opinion. This study was based on a small number of cases. We are planning to examine a large number of cases in the future. (Line 209-212)
“Our study suffers from the limitation that it was a single facility retrospective study of a small group of patients. Therefore, prospective randomized controlled trials with many cases are required to validate our findings. We are planning to examine prospective randomized controlled trials with a large number of cases in the future.”
Comments 2: Have the author considered to provide further image for the compare between thin stent and thick stent with photo in your figure (before deployment and after deployment)
Response 2: Thank you for valuable comment. Photos before and after stent deployment used in this study are shown. (Figure 2, 3)
Reviewer 2 Report
Comments and Suggestions for Authors
The study is interesting. The authors should comment that HGS is a difficult techniques that should be performed only by highly skilled endoscopists in high volume centers
The inclusion criteria are quite unclear. Were enrolled only patients after failed ERCP or also naive cases?
The authors should explain in the discussion that there are also several other techniques for biliary drainage, citing some of them (in this regard cite some studies on LAMS, for example PMID: 34339667)
Author Response
Comments 1: The study is interesting. The authors should comment that HGS is a difficult techniques that should be performed only by highly skilled endoscopists in high volume centers
Response 1: Thanks for valuable comment. As pointed out, EUS-HGS should be performed in high volume center with a technically sufficient endoscopist. We added the information you indicated in the text. ‘‘EUS-HGS is a highly technical procedure and should only be performed by highly skilled endoscopists at high volume centers.’’ (Line 39-40)
Comments 2: The inclusion criteria are quite unclear. Were enrolled only patients after failed ERCP or also naive cases?
Response 2: Thanks for important comment. In this study, ERCP failure cases are included. On the other hand, ERCP is performed for naïve cases in which ERCP is possible.
The criteria for performing EUS-HGS at our institution are described below in the text. ‘‘The main indications for EUS-HGS at our hospital are: it is difficult to reach the duodenal papilla due to reconstructive bowel or gastrointestinal stenosis, bile duct cannulation is difficult and cholangitis is poorly controlled with transpapillary drainage with ERCP alone. (Line 57-60)
Comments 3: The authors should explain in the discussion that there are also several other techniques for biliary drainage, citing some of them (in this regard cite some studies on LAMS, for example PMID: 34339667)
Response 3: Thanks for pointing that out. We have added to the text other techniques for biliary drainage in discussion paragraphs. (Line 202-208)
‘‘Lumen-apposing metal stent (LAMS) is used for bile duct drainage recently, which is convenient because it combines a stent and an electrocautery dilator in one unit [RimbaÅŸ M. et al. Gastrointest Endosc. 2015, 82, 405.]. Previous reports have shown that LAMS can be used safely during bile duct drainage [Mangiavillano, B. et al. Gastrointest Endosc. 2022, 95, 115-122.]. However, bleeding has been reported as an adverse event when using electrocautery dilator [Kunda, R. et al. Surg Endosc. 2016, 30, 5002-5008.] [Honjo M. et al. Endosc Ultrasound. 2018 Nov-Dec;7(6):376-382]. In this study, fistula dilation was not required for placement of a thin delivery stent. Thin delivery stents may be useful because they can be safely performed without the need for energization and balloon fistula dilation.’’
Round 2
Reviewer 2 Report
Comments and Suggestions for Authors
Thank you for your response